# ResGen: Residual Diffusion Model for LiDAR-based Point Cloud Generation

## Abstract

While significant progress has been made in 2D image generation, the generation of 3D point clouds is less explored. Existing occupancy generation methods usually suffer from resolution limitations, while range image based methods are limited to single-frame rotating-scanning LiDAR points. In this paper, we propose ResGen, a framework towards realistic LiDAR-based point cloud generation. Our method first generates coarse 3D structures and then refines them into high-fidelity point clouds. Specifically, we build a 3D Residual Diffusion Model for refinement. With a pilot study revealing the theoretical shortcomings in existing approaches, we model our diffusion process for a *residual* between the coarse and refined point cloud. ResGen preserves fine-grained details and demonstrates applicability to multi-frame accumulated LiDAR point clouds. Experiments demonstrate that ResGen achieves superior results both qualitatively and quantitatively. Code will be made publicly available.

## 1 Introduction

Diffusion models (Ho et al., 2020; Song et al., 2020; Rombach et al., 2022) have achieved remarkable success in 2D image generation. However, 3D LiDAR point cloud generation presents unique challenges due to the complicated 3D structures and the fundamentally different spatial properties compared to 2D images.

Early explorations (Luo & Hu, 2021; Vahdat et al., 2022; Zhou et al., 2021) attempted to apply diffusion models to continuous 3D coordinates. While effective for generating single-object point clouds, these approaches struggled with large and complex scenes. Subsequent works (Lee et al., 2023; Liu et al., 2024; Xiong et al., 2023; Lee et al., 2024) adopted discretization, such as modeling occupancy grids, to enable scene-level generation. However, these methods suffered from limited output fidelity due to the fixed and coarse granularity of grids.

A parallel line of research focused on LiDAR-specific generation (Caccia et al., 2019; Hu et al., 2024; Zyrianov et al., 2022; Ran et al., 2024). Utilizing the inherent structure of LiDAR range images, these approaches produce point clouds by back projection from generated range images. This projection is tightly coupled to rotating-scanning LiDAR hardware, restricting its generalization.

To overcome these limitations, we propose **ResGen**, a novel two-stage framework for high-fidelity LiDAR point cloud generation. ResGen decomposes the generation process into a coarse-to-fine pipeline. In the first stage, we utilize a latent diffusion model to generate density voxels that encode point cloud density. Coarse point clouds are then sampled from these voxels. In the second stage, a Residual Diffusion Model refines the coarse output by predicting point-wise residuals, thereby restoring high-resolution details without being constrained by voxel granularity. ResGen is neither restricted to single-frame LiDAR point clouds as range image based methods, nor limited to coarse structure as occupancy-based methods.

At the core of our refinement stage is the **Residual Diffusion Model**, which treats the coarse point cloud as a fixed condition (detailed in Figure 3) and learns to generate residuals that bridge the gap between the coarse and fine point clouds. As discussed in § 4.2.1, previous methods learn the refinement by adding Gaussian noise to the ground-truth, which implicitly assumes a Gaussian distribution for residuals. In contrast, we do not assume a predefined distribution and choose to directly learn this residual distribution. We also provide an experimental justification, suggesting

that the distribution is indeed not Gaussian. With our novel design, ResGen achieves superior results both qualitatively and quantitatively. Furthermore, we extend our model with a tailored training pipeline for text-conditioned generation.

In summary, our main contributions are threefold. (1) We propose **ResGen**, a framework towards realistic LiDAR-based point cloud generation. (2) We introduce a novel **Residual Diffusion Model** supported by theoretical analysis. (3) Our implementation outperforms existing methods in various tasks, demonstrating versatility and effectiveness.

## 2    RELATED WORK

**Object-level 3D point cloud Generation**    As one of the pioneering works in the field, researchers proposed a diffusion probabilistic model for 3D point clouds (Luo & Hu, 2021). Subsequent work PVD (Zhou et al., 2021) utilizes a hybrid representation of points and voxels to model a unified probabilistic formulation, while LION (Vahdat et al., 2022) builds diffusion models in hierarchical latent spaces. Inspired by the recent success of Transformer, PointARU (Meng et al., 2025) generates point clouds autoregressively based on a multi-scale VAE. Although these works managed to transfer diffusion models to 3D space and generate point clouds with continuous coordinates, they focus on object-level point cloud generation and struggle to generalize to large-scale scenes with rich details.

**Large-scale 3D Occupancy Generation**    3D generation for large-scale scenes is also the focus of many works. Building upon the idea of voxelization (Liu et al., 2019; Yang et al., 2018; Graham et al., 2018; Liu et al., 2021), researchers apply a discrete diffusion model (Austin et al., 2021) for scene scale (Lee et al., 2023; Liu et al., 2024) by either combining a VQ-VAE (Van Den Oord et al., 2017) with it or utilizing a pyramid structure. Additionally, SemCity (Lee et al., 2024) represents 3D scenes by triplanes to enable efficient generation, while UltraLiDAR (Xiong et al., 2023) applies VQ encoding for point cloud completion and a Transformer for generation. As these works tackle the large scale by voxelization, they are only capable of generating discrete occupancy grids.

**Range image based LiDAR Point Cloud Generation**    Following earlier attempts based on 2D point maps (Caccia et al., 2019), various methods (Hu et al., 2024; Zyrianov et al., 2022; Ran et al., 2024; Nakashima & Kurazume, 2021; Nakashima et al., 2023; Yan et al., 2025) specifically generate point clouds obtained by LiDAR based on range images. LiDARGen (Zyrianov et al., 2022) applies a score-based model, which is powerful but suffers from slow sampling speed. LiDM (Ran et al., 2024) builds an autoencoder for range images, adding geometric priors to generate LiDAR realistic scenes. Range image based methods heavily rely on the property of LiDAR. Therefore, they can only generate point clouds from single-frame LiDAR scans, and LiDAR parameters need to be specified for generation. In contrast, our method is applicable to more general point clouds, including single-frame and multi-frame LiDAR point clouds.

## 3    PRELIMINARY: DIFFUSION MODELS

To align with our subsequent theoretical derivation, we first briefly revisit the classical Denoising Diffusion Probabilistic Model (DDPM) (Ho et al., 2020), which introduces a forward "diffusion" process that gradually transforms a data sample into Gaussian noise, and a learnable "reversed" process that recovers data from noise.

With $\{\beta_t\}$ being a "variance schedule", the forward process can be formulated as

$$\mathbf{x}_t = \sqrt{\bar{\alpha}_t}\mathbf{x}_0 + \sqrt{1 - \bar{\alpha}_t}\boldsymbol{\epsilon}, \tag{1}$$

where $\boldsymbol{\epsilon} \sim \mathcal{N}(0, \mathbf{I})$, $\alpha_t = 1 - \beta_t$, and $\bar{\alpha}_t = \prod_{s=1}^{t} \alpha_s$. During training, the loss $\|\boldsymbol{\epsilon} - \boldsymbol{\epsilon}_\theta(\mathbf{x}_t, t)\|^2$ is minimized, with $\mathbf{x}_t$ defined in Eq. 1.

During inference, starting from $\mathbf{x}_T \sim \mathcal{N}(0, \mathbf{I})$, the model iteratively denoises to obtain the generated result $\mathbf{x}_0$, using the formula $\mathbf{x}_{t-1} = \frac{1}{\sqrt{\alpha_t}}\left(\mathbf{x}_t - \frac{1-\alpha_t}{\sqrt{1-\bar{\alpha}_t}}\boldsymbol{\epsilon}_\theta(\mathbf{x}_t, t)\right) + \sigma_t\mathbf{z}$.

The form of $\mathbf{x}$ varies by task: $\mathbf{x} \in \mathbb{R}^{H \times W}$ for **2D diffusion**, which is applied in our coarse generation module, and $\mathbf{x} \in \mathbb{R}^{N \times 3}$ for **3D point cloud diffusion**, which is used in our Residual Diffusion Model.

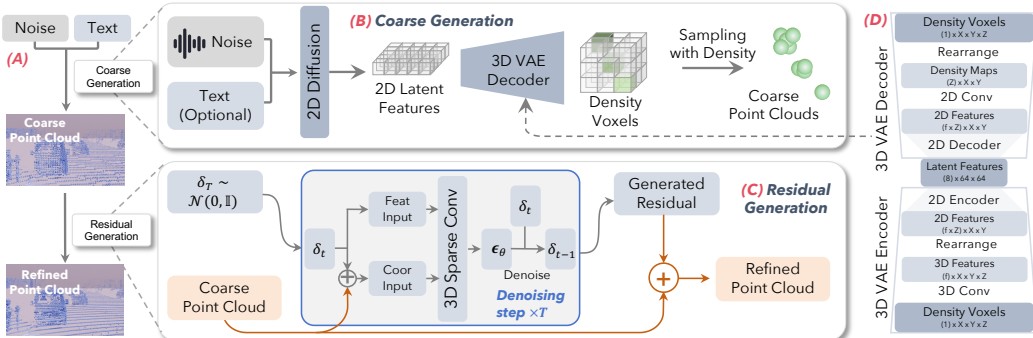

Figure 1: The main pipeline of our method. *(A)* Our method first generates coarse point clouds and then refines them into high-fidelity results. *(B)* We first generate 3D density voxels and then sample from these voxels to produce coarse point clouds (§ 4.1). *(C)* Coarse point clouds are refined by generating a residual between the target and them (§ 4.2). *(D)* The structure of our VAE for density voxels (§ 4.1.1). The "2D encoder" and "2D decoder" are constructed following LDM (Rombach et al., 2022). The data dimensionality at each processing step is illustrated in each block, with the number of features explicitly indicated in parentheses. (X, Y, Z) represents spatial dimensions, set as (512, 512, 64) in the experiment, with Z denoting the height.

## 4 METHOD

As described in § 1, our approach consists of a coarse generation stage and a refinement stage. The initial coarse generation stage operates in voxel space to ensure structural stability, while the refinement stage operates in continuous space to preserve quality.

### 4.1 COARSE GENERATION

The coarse generation stage integrates a 3D Variational Autoencoder (VAE) with a 2D diffusion model. The 3D VAE encodes the input into a 2D latent space, which is then processed by the 2D diffusion model.

#### 4.1.1 VAE FOR 3D DENSITY VOXELS

Our 3D Variational Autoencoder (VAE) operates on *3D density voxels* instead of directly processing point clouds. Unlike traditional occupancy grids, our 3D density voxel representation $V \in \mathbb{R}^{D \times H \times W}$ encodes point density rather than binary occupancy. Each voxel $V[i, j, k]$ stores an integer indicating the number of points that fall within that voxel.

Given point clouds, we compute the number of points falling within each voxel to obtain 3D density voxels. The VAE encoder then transforms 3D density voxels into 2D feature maps, while the decoder transforms 2D feature maps back to 3D density voxels. The overall structure of our VAE is shown in Figure 1. The information along the Z-dimension of 3D feature voxels is simply merged when transformed into 2D feature maps. This straightforward design is sufficient thanks to our coarse-to-fine strategy, where the coarse stage is only required to generate approximate shapes.

#### 4.1.2 VAE TRAINING

The fundamental training objective of our VAE combines a reconstruction loss with a Kullback–Leibler (KL) divergence term, where the reconstruction term is implemented as a Smooth L1 loss (Girshick, 2015) between voxel values. However, this objective formulates a continuous regression problem, where models generate ambiguous shapes in transition regions between occupied and empty voxels. Therefore, in addition to reconstructing the density values, the decoder also predicts a binary **occupancy grid**. This grid indicates whether each voxel is occupied (contains at least one point) or not. We derive the ground truth occupancy grid from the input density voxels and supervise the model's prediction using a Binary Cross-Entropy loss, denoted as $\mathcal{L}_{\text{BCE}}$. This provides a direct signal about shapes.

Furthermore, to better handle the class imbalance between occupied and empty voxels, we incorporate two additional regularization terms: the Dice Loss (Milletari et al., 2016) and the Focal Loss (Lin et al., 2017). Consequently, the overall training loss is formulated as $\mathcal{L} = \mathcal{L}_{\text{recon}} + \lambda\mathcal{L}_{\text{KL}} + \mathcal{L}_{\text{BCE}} + \mathcal{L}_{\text{dice}} + \mathcal{L}_{\text{focal}}$.

### 4.1.3 2D LATENT DIFFUSION

We follow the architecture of LDM (Rombach et al., 2022) to construct the 2D latent diffusion model. During training, we first use the encoder of the VAE discussed above to obtain 2D latent features. Noise is added to these latent features following Eq. 1 and the model learns to recover the features from noise. During inference, given a 2D noise input, the model generates 2D latent features, which are further decoded by the VAE decoder. For text-conditioned training, we obtain the textual descriptions through a Vision Language Model by analyzing the images corresponding to point clouds.

### 4.1.4 SAMPLING WITH DENSITY

2D latent features generated by the 2D latent diffusion model are decoded by the VAE decoder to yield 3D density voxels. These voxels enable direct point cloud sampling: $V[i, j, k]$ points are sampled at the voxel center within each voxel (with $V[i, j, k]$ defined in § 4.1.1), and the points sampled from all voxels are merged to form a coarse, continuous-valued 3D point cloud. This sampling approach is general as it requires no prior knowledge about the target point cloud's structure or distribution.

## 4.2 REFINEMENT BY RESIDUAL GENERATION

While the previous stage generates coarse point clouds, it loses local details and pattern characteristics (such as LiDAR scan beams). The refinement module aims to recover this information to produce a detailed point cloud that follows the expected distribution. We formulate this as a conditional generation problem: generating the target refined point cloud $\mathbf{x}_{\text{tar}} \in \mathbf{R}^{N \times 3}$ given the coarse point cloud $\mathbf{x}_{\text{pri}} \in \mathbf{R}^{N \times 3}$ as a prior sample. We call it as a prior rather than a condition, because it shares the same modality as the generation target and can be regarded as its primitive form. We first analyze the theoretical shortcomings of existing methods focusing on this problem through a pilot study, and then propose our novel Residual Diffusion Model for refinement.

### 4.2.1 PILOT STUDY: THEORETICAL GAP IN PREVIOUS METHODS

Previous methods have employed a special diffusion model for point cloud generation, which we call "Partial Diffusion Model", as it employs an *incomplete* diffusion process. Specifically, a small diffusion timestep $t$ is used for training such that the perturbed samples retain the underlying structure of the original data rather than degrading entirely into Gaussian noise. During generation, the model starts from the given prior sample (rather than pure Gaussian noise) and iteratively denoises it to produce the final output.

LiDiff (Nunes et al., 2024) is a representative that employs this approach, focusing on point cloud completion. Similar to our task, it refines a coarse point cloud (which is sparse) to produce the target (which is dense). During training, noise is added by



$$\mathbf{x}_t = \mathbf{x}_0 + \sqrt{1 - \bar{\alpha}_t}\boldsymbol{\epsilon}. \tag{2}$$

Given that $\sqrt{\bar{\alpha}_t}$ is close to 1 when $t$ is small, this formulation can be viewed as an approximation of Eq. 1.

With Eq. 2, LiDiff makes an assumption for the existence of a timestep $t$ such that $\mathbf{x}_{\text{pri}} =$

Figure 2: Empirical distribution of the residuals between our target point clouds and their corresponding coarse versions. Three subplots respectively depict the distributions of the three components of the residuals.

$\mathbf{x}_{\text{tar}} + \sqrt{1 - \bar{\alpha}_t}\boldsymbol{\epsilon}$, where $\mathbf{x}_{\text{pri}}$ is the prior coarse point cloud and $\mathbf{x}_{\text{tar}} = \mathbf{x}_0$ is the target of generation.

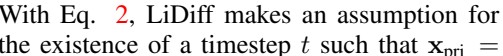

This equals $\mathbf{x}_{\text{pri}} - \mathbf{x}_{\text{tar}} = \sqrt{1 - \bar{\alpha}_t}\epsilon$, which means that *the difference between the coarse and target point cloud follows a Gaussian distribution.*

This assumption *does not hold* in many cases. Specifically, we performed Kolmogorov-Smirno hypothesis testing (Massey Jr, 1951) to examine the distribution in our scenario. Results yield statistically significant evidence ($p \leq 0.01$) to reject the assumption, which means that the residuals are statistically inconsistent with a Gaussian distribution. Furthermore, as visually confirmed in Figure 2, the empirical distribution of these residuals better approximates a uniform distribution. Details of the experiment are shown in Appendix B.

### 4.2.2 FORMULATION OF THE RESIDUAL DIFFUSION MODEL

The analysis in the pilot study reveals that partial diffusion models inherently assume a Gaussian distribution of the difference between $\mathbf{x}_{\text{pri}}$ and $\mathbf{x}_{\text{tar}}$, thus introducing theoretical limitations. This insight leads us to a new perspective: *Can we instead develop a diffusion model that directly learns this distribution?*

Building upon this idea, we propose a Residual Diffusion Model. Formally, we define the **residual** as

$$\boldsymbol{\delta} = \mathbf{x}_{\text{tar}} - \mathbf{x}_{\text{pri}}. \tag{3}$$

In our scenerio, this residual is computed by first voxelizing the target (where points are assigned to their respective voxels) and then measuring the difference between each point and its corresponding voxel center, as our coarse point clouds are obtained by sampling at voxel centers.

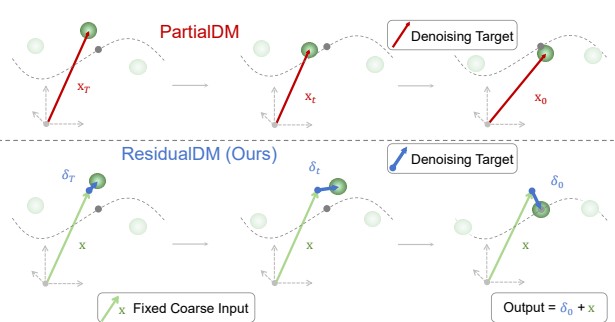

Figure 3: Illustration of our Residual Diffusion Model (Residual DM). Our model learns a *residual* between the coarse point cloud and the target, while the previous Partial DM directly learns the target.

As illustrated in Figure 3, instead of directly modeling the target distribution, this model focuses on learning the *residual* between the prior distribution and the target distribution. Specifically, we build a conditional diffusion model for residual $\boldsymbol{\delta}$ (defined in Eq. 3), with the reversed process $p_\theta(\boldsymbol{\delta}_{0:T} \mid \mathbf{x}_{\text{pri}}) := p(\boldsymbol{\delta}_T) \prod_{t=1}^{T} p_\theta(\boldsymbol{\delta}_{t-1} \mid \boldsymbol{\delta}_t, \mathbf{x}_{\text{pri}})$, where $\boldsymbol{\delta}_T \sim \mathcal{N}(0, \mathbf{I})$ and $\boldsymbol{\delta}_0$ follows the targeted residual distribution. $p_\theta$ is a normal distribution with variance $\sigma_t^2 \mathbf{I}$ and mean $\boldsymbol{\mu}_\theta = \frac{1}{\sqrt{\alpha_t}} \left( \boldsymbol{\delta}_t - \frac{1-\alpha_t}{\sqrt{1-\bar{\alpha}_t}} \boldsymbol{\epsilon}_\theta(\boldsymbol{\delta}_t, \mathbf{x}_{\text{pri}}, t) \right)$, with $\alpha_t$ and $\bar{\alpha}_t$ defined in § 3.

During training, GT point clouds in datasets are provided as the target. Their coarse versions are obtained by voxelization, following the discussion in the Pilot Study. In this way, we prepare data pairs of coarse and refined point clouds, where the points are naturally one-to-one corresponded by the voxelization operation. The residual $\boldsymbol{\delta}_0$ is then computed using Eq. 3. Different from Eq. 2, a random noise $\boldsymbol{\epsilon}$ is added to **the residual** by

$$\boldsymbol{\delta}_t = \sqrt{\bar{\alpha}_t}\boldsymbol{\delta}_0 + \sqrt{1 - \bar{\alpha}_t}\boldsymbol{\epsilon}, \tag{4}$$

and the loss $\|\boldsymbol{\epsilon} - \boldsymbol{\epsilon}_\theta(\boldsymbol{\delta}_t, \mathbf{x}_{\text{pri}}, t)\|^2$ is minimized.

During inference, conditioned on the coarse point cloud, $\boldsymbol{\delta}_0$ is generated from Gaussian noise by iteratively applying $\boldsymbol{\delta}_{t-1} = \frac{1}{\sqrt{\alpha_t}} \left( \boldsymbol{\delta}_t - \frac{1-\alpha_t}{\sqrt{1-\bar{\alpha}_t}} \boldsymbol{\epsilon}_\theta(\boldsymbol{\delta}_t, \mathbf{x}_{\text{pri}}, t) \right) + \sigma_t \mathbf{z}$. As shown in Figure 1, the refined point cloud is obtained by adding the generated residual $\boldsymbol{\delta}_0$ to the coarse point cloud.

Our method differs from previous Partial Diffusion Models (Partial DMs for short) in two aspects:

- Our method performs denoising adjustments on the residual, with the prior coordinates fixed as a condition, while Partial DMs perform denoising on the prior absolute coordinates.

- We establish a *complete* diffusion process specifically designed for residual modeling, while Partial DMs employ an incomplete process, as discussed in the pilot study.

With the differences above, our method demonstrates superior stability compared to previous methods. We also provide rigorous theoretical foundations, imposing no distributional assumptions on the target residuals.

### 4.2.3 IMPLEMENTATION

We implement the model $\epsilon_\theta(\boldsymbol{\delta}_t, \mathbf{x}_{\mathrm{pri}}, t)$ using a MinkowskiNet (Choy et al., 2019) based on 3D sparse convolutions. The network operates directly on 3D point clouds, accepting two inputs for each sample: *the coordinates of each point* and *the corresponding point features*. As shown in Figure 1, the coordinate input is obtained by adding the current residual values $\boldsymbol{\delta}_t$ to the prior coordinates $\mathbf{x}_{\mathrm{pri}}$, obtaining the current absolute coordinates. In this way, prior conditions are injected into the model. For feature input, we explore two variants: using the residual directly as features (as shown in Figure 1), and concatenating the residual with coordinates (formulated as $concat[\boldsymbol{\delta}_t, \boldsymbol{\delta}_t + \mathbf{x}_{\mathrm{pri}}]$).

## 5 EXPERIMENTS

Our experiments consist of multiple components, demonstrating both the versatility and effectiveness of our method.

### 5.1 IMPLEMENTAION

We conduct experiments for LiDAR generation on the KITTI-360 dataset (Liao et al., 2023), reserving sequence 03 for validation as per previous works. For coarse generation, the VAE is trained for 70k steps and the 2D diffusion model for 230k steps. The Residual Diffusion Model is trained for 6k steps. During inference, we use 50 denoising steps for 2D diffusion and 10 steps for the 3D refinement. Appendix C.2 lists more details.

For quantitative evaluation, we use Jensen-Shannon Divergence (JSD) and Minimum Matching Distance (MMD) (Achlioptas et al., 2018) to measure the statistical similarity between the generated and real point cloud distributions. The evaluation is bounded within a 50-meter square along the ground plane. 2000 samples are generated for the evaluation of each method. Please refer to Appendix C.3 for more details.

### 5.2 UNCONDITIONAL LIDAR SCAN GENERATION

Table 1: Comparison of unconditional LiDAR Scan Generation with previous methods. "↓": lower is better. REAP indicates the relative error between the average point counts of generated samples and real-world samples. User preference indicates the percentage of users who considered the results among the best. Please refer to Appendix C.3 for more details of the evaluation.

| Method | JSD ↓ | MMD ↓ $(10^{-2})$ | REAP↓ (%) | USER PREFERENCE ↑(%) |
|---|---|---|---|---|
| LiDM (Ran et al., 2024) | 0.439 | 0.256 | 57.81 | 40.40 |
| UltraLiDAR (Xiong et al., 2023) | 0.787 | 0.490 | 71.33 | 61.15 |
| LiDARVAE (Caccia et al., 2019) | 0.438 | 0.297 | 53.16 | 14.6 |
| LiDARGAN (Caccia et al., 2019) | 0.469 | 0.543 | 73.97 | 10.40 |
| LiDARGen (Zyrianov et al., 2022) | 0.413 | 0.225 | 58.24 | 54.60 |
| ProjectedGAN (Sauer et al., 2021) | 0.425 | 0.213 | 58.67 | 33.85 |
| **ResGen (Ours)** | **0.182** | **0.206** | **1.63** | **85.00** |

**Superior quantitative results**   Our approach demonstrates superior quantitative performance in both JSD and MMD. We also conducted a user study to provide subjective assessments, where ours was perceived as the most realistic by participants. Details are shown in Table 1.

**Better density realism**   Beyond shapes, we argue that point density is also a crucial dimension for evaluating realism. Our adaptive sampling strategy ensures that the generated point clouds maintain density characteristics statistically consistent with real-world scans—a capability beyond the reach of either range-image-based methods (which inherit fixed angular resolution) or discretization approaches (suffering from voxel-induced uniformity). We present a metric (REAP) in Table 1, where our approach outperforms all other methods by a significant margin.

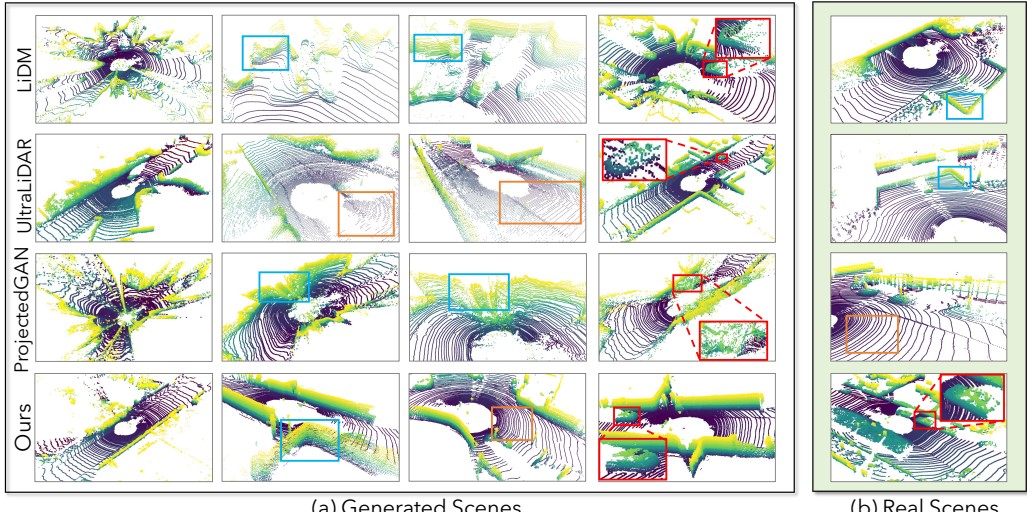

(a) Generated Scenes        (b) Real Scenes

Figure 4: Qualitative comparisons for unconditional single-frame LiDAR point cloud generation. Key structural elements are highlighted in distinct border colors (walls:blue, ground:orange, vehicles:red). The rightmost column provides real data for reference to the desired effect of generation. Our results exhibit clear geometry structures, including straight walls and realistic cars, and preserve scanning patterns. In contrast, LiDM generates over-smooth patterns, curved walls, and incomplete cars, while UltraLiDAR ruins the scanning pattern due to discretization.

**Realistic structures and patterns** As shown in Figure 4, our results demonstrate realistic 3D structures and preserves LiDAR scanning patterns. In contrast, range image-based methods (e.g. LiDM and ProjectedGAN), which forcibly project 3D structures into 2D space, exhibit inherent limitations in representing 3D features. UltraLiDAR, being designed for discrete voxel generation, suffers from resolution limitations. To the best of our knowledge, our method is **the first** to automatically generate realistic LiDAR scanning patterns without manually applying LiDAR sensor-specific priors (such as the fixed angular grid inherent to range-image representations).

**More realistic layouts** Our method overcomes a typical drawback of range-image-based methods. As shown in the first column of Figure 4, LiDM and ProjectedGAN generate unrealistic multi-way intersections—a scenario that rarely exists in real-world data. We argue that this indicates the fundamental limitation of range image based methods: they only comprehend 2D appearances in the range view rather than holistically modeling 3D layouts. Therefore, these methods are prone to generating repetitive 2D features, which assemble into unrealistic layouts of multi-way intersections when projected into 3D space.

## 5.3 CONDITIONAL LiDAR SCAN GENERATION

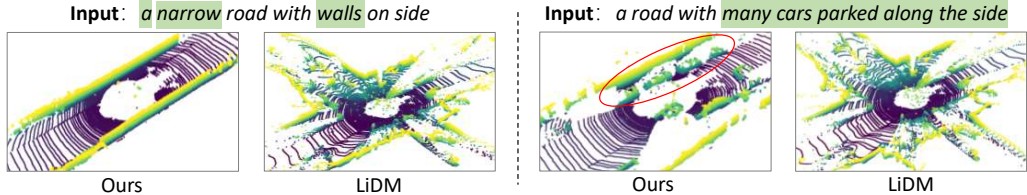

Figure 5: Conditional generation results. Only LiDM is compared because other counterparts do not support text-condition. Our method correctly responds to conditions, including the road count, road width, roadside objects, and vehicles. LiDM generates multiple roads instead of following the command of "a"; it fails to generate straight walls; it only generates a car, instead of "many cars".

In this section, we demonstrate text-conditioned generation capabilities of ResGen. As the KITTI-360 dateset lack text annotations for training, we complement them through a Vision Language

Model (VLM) by analyzing the corresponding images offered by the dataset. Qwen2.5-VL is applied as the VLM, with prompts in Appendix C.2.3.

We qualitatively compare our method against LiDM. As shown in Figure 5, our approach demonstrates superior control, correctly responding to textual conditions regarding road count, road width, and the presence of roadside objects and vehicles. In contrast, LiDM fails to satisfy most control requirements. These results indicate that our method achieves enhanced controllability and produces more realistic generation outputs.

## 5.4 MULTI-FRAME ACCUMULATED POINT CLOUD GENERATION

In this section, we conduct an additional experiment beyond generating single-frame LiDAR-scanned scenes to demonstrate our method's capability towards general point cloud generation. We focus on multi-frame accumulated LiDAR point clouds, which serve as the target in the completion task addressed by LiDiff. We follow LiDiff to prepare the data, where training samples are created by aggregating scans to form a static map, cropping a region corresponding to an original scan, and randomly sampling 180k points ("Real Scenes" in Figure 6 for visualization). We use the SemanticKITTI dataset (Behley et al., 2019), also following LiDiff.

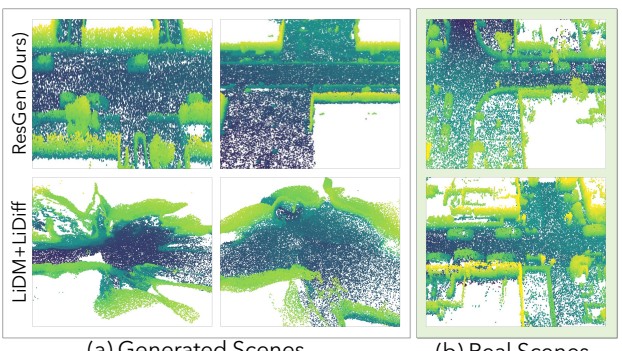

(a) Generated Scenes      (b) Real Scenes

Figure 6: Qualitative comparisons for accumulated point cloud generation. Our results exhibit much higher quality than the other. Please kindly note that other methods in Figure 4 *cannot* produce such accumulated point clouds.

Additional implementation details can be found in Appendix C.1 and C.2.

**Compared Methods and Results** Existing range image based methods *cannot* generate multi-frame accumulated point clouds. We employ the LiDAR completion model LiDiff to process the outputs of LiDM, enabling a fair comparative analysis. As shown in Table 2, our approach surpasses this counterpart by a large margin. Figure 6 presents a qualitative comparison, where our results exhibit coherent 3D geometric structures, clearly distinguishable objects, and diversity. In contrast, the competing results appear ambiguous, with poor diversity.

Table 2: Quantitative comparison of multi-frame accumulated point cloud generation. "↓": lower is better. UP represents user preference, indicating a percentage, defined as in Table 1. For LiDM+LiDiff, LiDM generates LiDAR scans, and LiDiff completes them.

| Method | JSD ↓ | MMD ↓ $(10^{-2})$ | UP ↑(%) |
|---|---|---|---|
| LiDM | 0.559 | 0.263 | / |
| LiDM + LiDiff | 0.217 | 0.197 | 0 |
| **ResGen (ours)** | **0.119** | **0.140** | **100** |

## 5.5 ABLATION STUDY

**Residual DM versus Partial DM** We first conduct ablation experiments to validate the effectiveness of our Residual Diffusion Model.

For comparison, we implement a partial diffusion model (Partial DM) by adapting LiDiff to our task. We empirically discover that removing the "per-point guidance" structure in the original model yields better performance for our task. Therefore, this modified implementation is adopted as our baseline.

Table 3: Ablation studies for the Residual Diffusion Model. Implementation variants of our Residual DM achieved similar results, both outperforming the Partial DM counterpart. For the definition of our two variants, please refer to § 4.2.3.

| | | Single Scan | | Accumulated | |
|---|---|---|---|---|---|
| **Method** | | JSD ↓ | MMD ↓ $(10^{-2})$ | JSD ↓ | MMD ↓ $(10^{-2})$ |
| Partial DM (LiDiff) | | 0.250 | **0.206** | 0.365 | **0.140** |
| **Residual DM** | Concatenated | **0.182** | **0.206** | 0.135 | **0.140** |
| **(Ours)** | Residual only | 0.186 | **0.206** | **0.119** | **0.140** |

As shown in Table 3, when applied as the refinement module, *our Residual Diffusion Model demonstrates superior quantitative results compared to PartialDM in both experiments.*

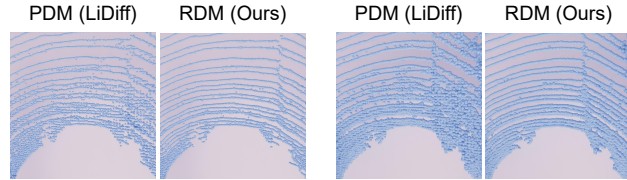

Figure 7: Typical examples of comparisons between our Residual DM and Lidiff's Partial DM. The latter degrades the scanning patterns and leads to more outlier artifacts.

Moreover, as shown in Figure 7, PartialDM produces more artifacts of disordered points than ours. We argue that this phenomenon likely occurs because these points reside in the tails of the target-prior residual distribution— regions where the probability density is extremely low under a Gaussian assumption. Since partial diffusion models forcibly impose a Gaussian distribution assumption, they inherently fail to properly model these outlier points.

**Residual DM implementation variants**    As also shown in Table 3, two implementation variants of our Residual DM (mentioned in § 4.2.3) achieve similar performance. This demonstrates that *the residual alone is sufficient for modeling the generation process*, and incorporating absolute coordinates provides few additional benefits.

**VAE design**    To evaluate the effectiveness of our VAE design, we conduct reconstruction experiments on the validation set, using Chamfer Distance (CD) and Earth Mover's Distance (EMD) as evaluation metrics (Achlioptas et al., 2018). As shown in Table 4, the performance degrades significantly when the regularization loss is removed. Notably, removing the occupancy loss leads to an even more drastic drop in performance. This aligns with our discussion in Section 4.1, where we emphasized that the occupancy loss $\mathcal{L}_{\text{BCE}}$ provides a strong and direct shape supervision signal, which is crucial for stable VAE training.

Table 4: Ablation studies on different VAE loss components. Removing the occupancy loss (*w/o occloss*) leads to a substantial drop, highlighting its critical role in shape learning.

| | Single Scan | | Accumulated | |
|---|---|---|---|---|
| **Method** | CD ↓ | EMD ↓ | CD ↓ | EMD ↓ |
| **Ours** | **0.050** | **0.130** | **0.076** | **0.219** |
| w/o regloss | 0.129 | 0.221 | 0.133 | 0.269 |
| w/o occloss | 6.812 | 2.662 | 4.574 | 1.277 |

**Ablation for the overall pipeline** Preceding experiments all incorporate the refinement module, focusing on exploring its various design configurations. To further validate the effectiveness of our overall two-stage framework, we design additional experiments. As shown in Table 5, performance degrades significantly when the refinement module is omitted. Furthermore, we observe additional performance drop when further disabling the regularization loss (Dice and Focal loss) in the VAE, aligning with the results in Table 4.

Table 5: Ablation studies that demonstrate the overall effectiveness of our model. Disabling the refinement module yields significant performance degradation, with an additional drop when further ablating the Dice and Focal loss for VAE regularization.

| | Single Scan | | Accumulated | |
|---|---|---|---|---|
| **Method** | JSD ↓ | MMD ↓ ($10^{-2}$) | JSD ↓ | MMD ↓ ($10^{-2}$) |
| **ResGen** | **0.182** | **0.206** | **0.119** | **0.140** |
| w/o refinement | 0.737 | **0.206** | 0.780 | 0.142 |
| w/o regloss | 0.737 | 0.209 | 0.780 | 0.152 |

## 6  CONCLUSION

In this paper, we propose **ResGen** for generating LiDAR-based point clouds. In our generation pipeline, the Residual Diffusion Model is proposed, based on the analysis of the theoretical shortcomings in previous methods. Our framework can generate both single-frame and multi-frame accumulated LiDAR point clouds, demonstrating both the versatility and effectiveness. We anticipate that our method will offer a different perspective for both point cloud generation tasks and conditional diffusion models.

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

## A  DEMONSTRATION

We include a **video demonstration** as the supplementary material. The video provides an overview of the proposed method and presents qualitative comparisons with baseline methods. We encourage you to watch the video for a more intuitive and comprehensive understanding of our method.

## B  PILOT STUDY DETAILS

### B.1  RESIDUAL CALCULATION

Our target is the residual between the target and the prior sample. Note that our prior coarse point cloud is obtained by sampling at voxel centers, which can be regarded as a voxelized version of the target point cloud. Therefore, the residual is computed by first voxelizing the target (where points are assigned to their respective voxels) and then measuring the difference between each point and its corresponding voxel center. We calculate residuals across all grid cells of 10 point cloud samples. We find alternative sampling strategies (e.g., analyzing 100 samples with 10% random downsampling per grid cell) yield consistent results.

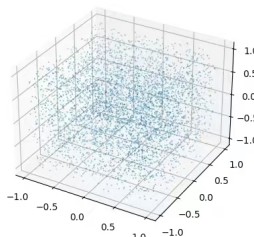

Figure 8: Visualization of the residual between the target and the prior sample.

### B.2  HYPOTHESIS TESTING

Hypothesis testing can be employed to reject the assumption that samples follow a particular distribution. The procedure involves: (1) formulating the null hypothesis that the sample is drawn from the target distribution, (2) computing the p-value, and (3) rejecting the hypothesis if the p-value falls below a predetermined threshold (typically 0.05).

We initially assumed that the residuals followed a normal distribution. We parameterized a normal distribution using the mean and standard deviation of residual samples and performed the Kolmogorov-Smirnov test, obtaining p-value = 0, rejecting normality. We further investigated truncated normal distributions by fitting optimal parameters to the samples, obtaining p = 0.0001 (still rejecting the hypothesis). These results demonstrate that the residuals do not follow a Gaussian distribution.

## C  MORE IMPLEMENTATION DETAILS

### C.1  DATA PREPARATION AND PREPROCESSING

**Datasets**  We evaluate LiDAR point cloud generation on the **KITTI-360** dataset and multi-frame accumulated point cloud generation on **SemanticKITTI**.

**Training/Validation Split**  Following previous works, all models are trained using the official training sets. One sequence is reserved from each dataset for validation: **sequence 03** in KITTI-360 and **sequence 08** in SemanticKITTI. All evaluations are conducted on the validation sequences.

**Data for Multi-frame accumulated Point Cloud Generation**  Following LiDiff, we utilize the poses in the SemanticKITTI dataset to aggregate all scans within each sequence and construct a static map by removing moving objects based on semantic labels. For each original scan frame, we crop the corresponding region ofhe map according to its position and range, and then randomly sample 180,000 points to form a sample.

### C.2 TRAINING AND INFERENCE CONFIGURATION

#### C.2.1 COARSE GENERATION

**VAE Training Steps** 70k steps on a mixture of KITTI-360 and SemanticKITTI for LiDAR scan generation, and 62k steps on SemanticKITTI for multi-frame accumulated point cloud generation.

**2D Latent Diffusion Training Steps** 230k steps for KITTI-360 and 80k steps for SemanticKITTI.

**Inference Steps** 50 denoising steps are used for 2D diffusion during inference.

**Batch Size** 8 samples per batch.

**Optimizer** Adam optimizer is used with a learning rate of $1 \times 10^{-4}$ for the VAE and $5 \times 10^{-5}$ for the diffusion model.

#### C.2.2 REFINEMENT BY RESIDUAL GENERATION

**Training Steps** The residual diffusion model is trained for 6k steps.

**Inference Steps** 10 denoising steps are used for 3D diffusion during inference.

**Batch Size** 2 samples per batch.

**Optimizer** Adam optimizer with a learning rate of $1 \times 10^{-4}$.

#### C.2.3 TEXT-CONDITIONED GENERATION

**Prompt** We utilize the following prompt to analyze the images corresponding to point clouds, generate captions:

"This is a photo taken from a moving car. The photo shows the road on which the vehicle is driving. Is the road wide or narrow? What vehicles are on the road or beside the road? Are there buildings, trees or pedestrians? How many vehicles and pedestrians are on the road? Your answer should be short and no more than five sentences."

**VAE Training Steps** 70k steps on a mixture of KITTI-360 and SemanticKITTI.

**2D Latent Diffusion Training Steps** 80k steps on KITTI-360 only.

**Inference Steps** 50 denoising steps.

**Batch Size** 8 samples per batch.

**Optimizer** Adam optimizer is used with a learning rate of $1 \times 10^{-4}$ for the VAE and $5 \times 10^{-5}$ for the diffusion model.

### C.3 EVALUATION METRICS

We adopt both standard quantitative metrics and user-centric subjective evaluations to comprehensively assess the quality of generated point clouds.

For **quantitative evaluation**, we employ the commonly used **Jensen-Shannon Divergence (JSD)** and **Minimum Matching Distance (MMD)** (Achlioptas et al., 2018; Ran et al., 2024) to measure distribution-level similarity, and propose a customized metric called **Relative Error of Average number of Points (REAP)** to capture point density accuracy. For **paired reconstruction quality**, we adopt **Chamfer Distance (CD)** and **Earth Mover's Distance (EMD)**.

For **subjective evaluation**, we conduct a **User Study** and report **User Preference (UP)** scores.

**Jensen-Shannon Divergence (JSD)** JSD evaluates the occupancy similarity between the generated point cloud set $S$ and the reference set $R$ in voxelized space. Formally, it is defined as:

$$\text{JSD}\left(P_S \| P_R\right) = \frac{1}{2} D_{KL}\left(P_R \| M\right) + \frac{1}{2} D_{KL}\left(P_S \| M\right), \tag{5}$$

where $M = \frac{1}{2}(P_R + P_S)$ is the average distribution and $D_{KL}$ denotes the Kullback-Leibler divergence (Kullback & Leibler, 1951). A lower JSD indicates a closer match between the two distribu-

tions. Unlike LiDM (Ran et al., 2024), a smaller voxel size of 0.03 is adopted in our experiments to better measure fine-grained details.

**Minimum Matching Distance (MMD)**   MMD measures the average best-case similarity between point sets in the generated distribution $P_S$ and reference distribution $P_R$ using a pairwise distance metric (e.g., CD). It is defined as:

$$\text{MMD}\,(P_S \| P_R) = \frac{1}{|P_R|} \sum_{Y \in P_R} \min_{X \in P_S} D_{CD}(X, Y), \tag{6}$$

where $D_{CD}$ denotes the Chamfer Distance between point sets $X$ and $Y$. Unlike LiDM (Ran et al., 2024), which adopts a 2D projected version of MMD, we compute MMD in 3D space, making it more appropriate for measuring geometric similarity between point clouds. A voxel size of 0.5 is adopted.

**Chamfer Distance (CD)**   Chamfer Distance computes the average nearest-neighbor distance between two point sets $X$ and $Y$. It is defined as:

$$\text{CD}(X, Y) = \frac{1}{|X|} \sum_{x \in X} \min_{y \in Y} \|x - y\|_2^2 + \frac{1}{|Y|} \sum_{y \in Y} \min_{x \in X} \|y - x\|_2^2. \tag{7}$$

A smaller CD indicates a better alignment between the generated and reference shapes.

**Earth Mover's Distance (EMD)**   EMD measures the minimum cost of transforming one point set into another, assuming equal cardinality and mass. It is computed as the optimal bijective matching:

$$\text{EMD}(X, Y) = \min_{\phi: X \to Y} \frac{1}{|X|} \sum_{x \in X} \|x - \phi(x)\|_2, \tag{8}$$

where $\phi$ is a bijection between $X$ and $Y$. EMD is particularly suitable for evaluating fine-grained shape similarity.

**Relative Error of Average number of Points (REAP)**   REAP quantifies the density mismatch between generated and real point clouds. It is defined as the relative error in the average number of points per scan:

$$\text{REAP} = \frac{|\bar{N}_{\text{gen}} - \bar{N}_{\text{real}}|}{\bar{N}_{\text{real}}}, \tag{9}$$

where $\bar{N}_{\text{gen}}$ and $\bar{N}_{\text{real}}$ denote the average number of points in generated and real samples, respectively. A lower REAP value indicates better alignment with real-world scan densities.

**User Preference (UP)**   We design a user study to evaluate the perceptual quality of generated point clouds. For each experiment, 20 questions are constructed:

- **Single-frame LiDAR Generation:** Participants are shown three results and asked to select the most realistic one.
- **Multi-frame Accumulated Generation:** Participants are presented with eight candidates and asked to select their top three preferences.

User choices are aggregated and normalized to a $[0, 1]$ range to compute the final UP score. Higher UP values indicate stronger human preference for the method's outputs.

# D   MORE RESULTS

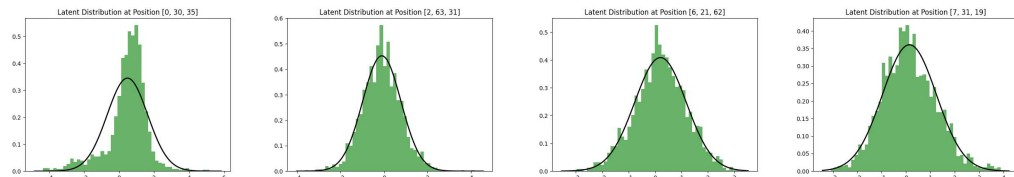

Figure 9: Examples of the per-dimension distribution of the latent space obtained by our 3D VAE in the first stage.The green histogram represents the empirical distribution, and the black curve represents the normal distribution fitted using the mean and variance of the green distribution.

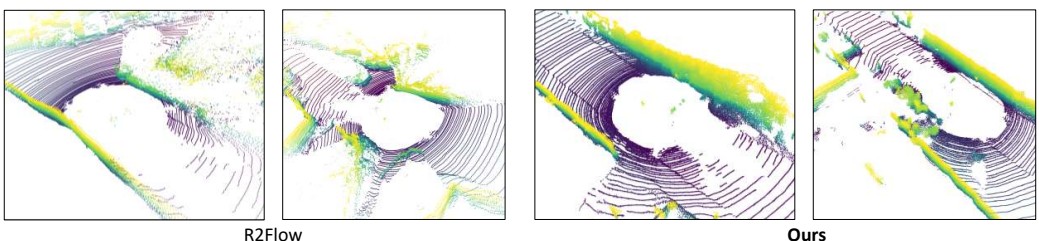

Figure 10: Visual comparisons between R2Flow and ours. R2Flow suffers from unstable structures and unclear objects. In contrast, our method generates stable structures and reasonable layouts.

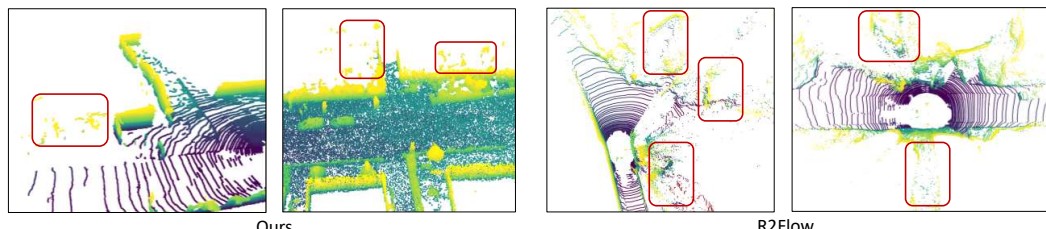

Figure 11: Limitation of our generation method. It occasionally produces some blurred outliers or noise points. However, our method has already shown improvement in this aspect compared to others.

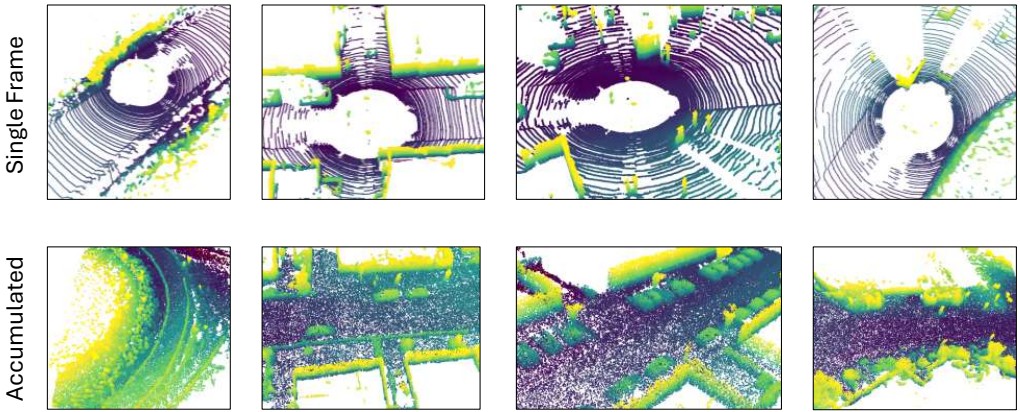

Figure 12: Our method demonstrates good diversity, generating scenes that encompass various types of roads.

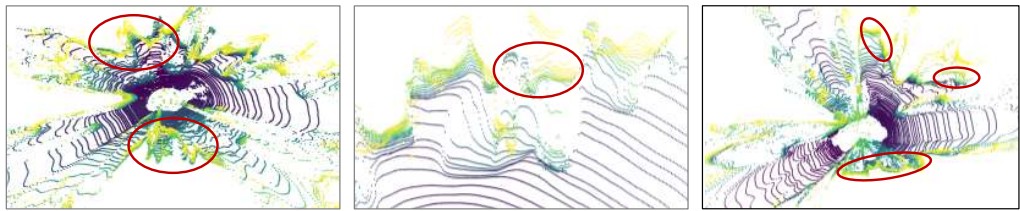

Figure 13: The characteristic of flying blurred edges produced by LiDM.

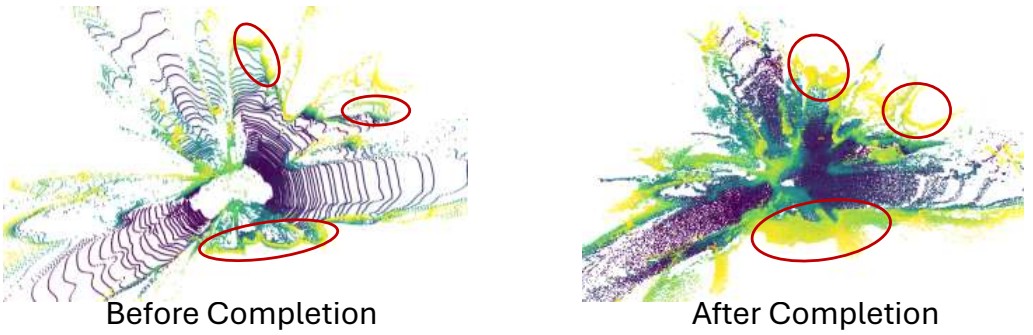

Before Completion                                        After Completion

Figure 14: LiDiff amplifies the visual effect of "blurred edges" produced by LiDM, by densifying the flying edge points.

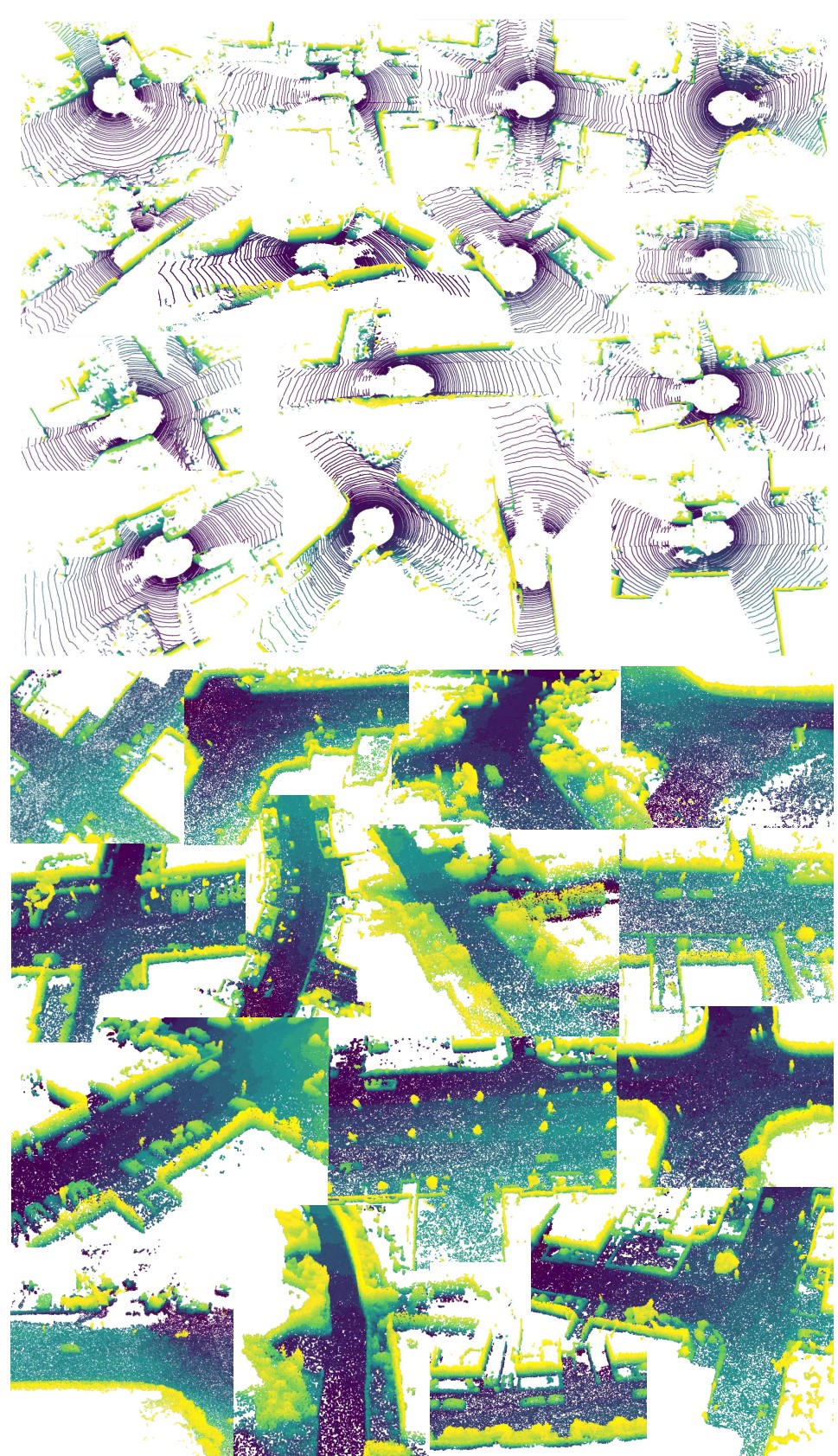

Figure 15: More visualizations of samples generated by our method.

