# OpenReview forum: "ResGen: Residual Diffusion Model for LiDAR-based Point Cloud Generation"
_ICLR.cc/2026/Conference — Submitted to ICLR 2026_

### Official Review · Reviewer_Qfi7 · 2025-11-01

**Soundness:** 3
**Presentation:** 3
**Contribution:** 3
**Rating:** 6
**Confidence:** 3

**Summary:**

This work proposes for 3D point cloud generation in the context of LiDAR scanning. The proposed method takes a two-step approach. In the first step, it generates density voxels in the 3D space. Then in the second step, it generates point clouds from density voxels. In the second step, points are sampled proportionally to the voxel density (not sure), and their locations are decided by their positions relative to voxel centers. The proposed method can also take text specifications as the condition, so its generation behavior can be controlled by text. The method is also extended multi-frame generation. The experiment shows that the method outperforms the baseline.

**Strengths:**

The two-step generation procedure is novel. The first step generates the general structure with the voxel structure. This design probably will ease the model's effort in deciding the general layout of shapes. The second step can then focus on local details.

The performance of the proposed model is better than baseline models.

**Weaknesses:**

The first step of generating voxels in the latent space is largely known in the field of image generation. Therefore, it somewhat discounts the novelty of the proposed work.

The writing of the work can be greatly improved. Here are a list of questions not clear from the writing.
1. Line 174, are there corresponding images of point clouds? Then, the caption of the image from the VLM is used as the caption of the point cloud? Is this really reliable as the image and point clouds are very different modelity.

2. The context of the pilot study is not clear. The LiDiff paper seems to use a diffusion model to generate target point cloud conditioned on a sparse one. Does the model have the Gaussian assumption in the **generation** stage? Figure 2 is hard to read. Are there curves? Or does the colored area represent the distribution?

3.  The inference stage is not clearly described. Given the voxel densities, the method will sample points according to these densities and then denoise these points all together in the single denoising model, right?

4. It is hard to understand how the method is applied to multi-frame generation. Do you treat these frames independently?

**Questions:**

In the user study, what measures do you take to avoid the bias? For example, how are these users hired? Is the evaluation double-blind? A lot of necessary details are missing.

It would be interesting to check the performance on voxel density as well. On the test set and point clouds generated by a competing method, you can compute their respective voxel densities. If you can check the performance of voxel densities, it is another method to understand where the improvement is from.

---

> ### Author Response · Authors · 2025-11-21
> **Response to Reviewer Qfi7 (Part 1/2)**
>
> We sincerely appreciate your positive feedback and detailed response and comments. We value your feedback greatly and try to resolve your concerns below.
>
> ### *W1: Generating voxels in the latent space is largely known*
>
> Thank you for your comment. Our decision to employ the latent space approach in our method was primarily driven by its proven stability, which ensures reliable generation of stable geometric structures in the first step. Moreover, we humbly emphasize that instead of simple voxels, **our method generates voxels with density, which can essentially be viewed as a density field.** This design enables it to control the sampling quantity and adapt to the non-uniform nature of point clouds.
>
> We greatly appreciate your suggestion, as it has led us to consider a new direction: whether we can adopt an encoding scheme more suited to our 3D scenarios to achieve improvements in quality. We plan to pursue this as a focus for our future work.
>
> ### *W2 Improvement of writing*
>
> Thank you very much for your valuable suggestions. They are extremely specific, detailed, and helpful. We will clarify the confusing points below, and revise them in a future version.
>
> ### W2.1: Corresponding images of point clouds and their reliability
>
> We apologize for not explaining this clearly in the text. **The "image" we refer to is not an image of the point cloud itself**, but an RGB photo of the real-world scene corresponding to that point cloud. Since using synchronized multi-modal sensors is standard practice in driving scene data collection, this photo is readily available. We feed this image to a VLM to obtain a caption that corresponds to the point cloud.
>
> **Reliability of the caption.** Although point clouds and images are different modalities, they correspond to the same real-world scenario. Therefore, many descriptive properties of the point cloud—such as the presence, size, and relative positional relationships of vehicles and objects—can be recognized from the image. We consider that these capabilities are primary for text-controlled generation. At the same time, we apply specific prompts to avoid the caption of attributes such as colors that cannot be recognized from the point cloud.
>
> ### W2.2: Gaussian assumption of LiDiff in the generation stage
>
> **Yes**, LiDiff uses a Gaussian assumption in the **generation** process. This can be derived from the fact that the generation process corresponds to the training process in a DDPM. Simply speaking, the X_T in training should have the same distribution as X_T in generation. As for LiDiff, since all X_T obtained in training is derived by adding Gaussian noise to the target, the starting point X_T for generation should also be the target plus Gaussian noise. **The purpose of our pilot study was to demonstrate that this assumption does not hold for real-world point clouds in our scenarios.** We hope the explanation above clarifies the context of our pilot study more clearly.
>
> **Explanation for Figure 2**
>
> In Figure 2, the colored areas, which show histograms, represent the distribution, while the curve in each subplot is just used to fit the upper bound of the histogram. We provide additional visualization of the distribution in Figure 8 of the revised paper, where residuals are scattered in 3D space.
>
> ### W2.3: Inference stage
>
> **We appreciate your correct understanding.** Given voxel densities, our method will sample points according to these densities and then denoise these points all together in a single denoising model, which is our Residual Diffusion Model for 3D point clouds.
>
> **Further Discussion: Importance of the sampling.** We would like to add that the sampling process is precisely a key link connecting our two stages. Without sampling with density, each voxel could only be processed into a fixed number of points, thereby losing the ability to adaptively control the number of generated points.
>
> ### W2.4: Multi-frame generation
>
> Instead of treating frames independently, **we first accumulate the multi-frame point cloud and then use this aggregation of points to train our model**. Visualizations of this aggregation are shown in Figure 6 (b) of the paper. We can directly use this aggregated point cloud as our training target because our method, built upon voxel and point representations, is inherently generalizable. It processes both single-frame and multi-frame point clouds under the same framework. In contrast, range image based methods are constrained to generating point clouds frame by frame, which is a limitation of such approaches.

---

> ### Author Response · Authors · 2025-11-21
> **Response to Reviewer Qfi7 (Part 2/2)**
>
> ### Q1: User Study
>
> Thank you for pointing this out, and we will clarify the details in the revision.
>
> We collected 20 completed questionnaires. All respondents are current **Master’s or PhD students in the same discipline, with background knowledge** in artificial intelligence, generative models, and 3D point clouds. In other words, the raters are not lay users, but have the technical expertise needed to make informed judgments about the plausibility and quality of LiDAR point clouds.
>
> The questionnaires were distributed via an **anonymous online survey platform** through a link. The platform does not record names or personal identifiers, and participants filled out the questionnaires independently, which helps reduce social and experimenter bias. Each questionnaire consists of 20 questions, 10 targeting the overall point cloud and 10 focusing on local regions of the point cloud. Each question consists of visualizations of point clouds from each method. For each method, we **randomly sample** generated point clouds. The sampled point clouds from different methods are mixed in the questionnaire and participants are **not informed** of the specific model that produces each sample. Therefore, from the participants’ perspective, the evaluation is **blind with respect to the compared methods**.
>
> While this is not a very large-scale crowd-sourcing study, we believe the combination of expert raters, anonymous responses, and randomly selected samples makes the user study a reasonable and informative complement to our quantitative metrics. Moreover, we observed that the results tend to stabilize after collecting approximately 10 questionnaires, so we think that the current scale of participants is sufficient.
>
> ### Q2: Performance on voxel density
>
> We sincerely thank you for this insightful suggestion, which offers a novel and helpful perspective to analyze our method. Following your comment, we conducted an additional evaluation directly for the **voxel density**. Concretely, we convert both the test set and point clouds generated by each method **(including ours)** into a 3D voxel-density grid of size $512\times512\times64$ with voxel size 0.1 m, and compute the JSD following Achlioptas et al. [1].
>
> | **Method** | **JSD (voxel)** | JSD (points) |
> | --- | --- | --- |
> | LiDARVAE | 0.46 | 0.44 |
> | UltraLiDAR | 0.71 | 0.79 |
> | LiDARGAN | 0.63 | 0.47 |
> | ProjectedGAN | 0.44 | 0.43 |
> | LiDM | 0.48 | 0.44 |
> | LiDARGen | **0.39** | 0.41 |
> | **Ours** | **0.39** | **0.18** |
> | Ours (w/o density sampling) | 0.43 | 0.20 |
>
> The lower the JSD, the better. As shown in the table above, although our method remains one of the top performers, the performance gap compared to other methods is significantly narrower than when evaluated using the final point cloud. This further demonstrates the effectiveness and necessity of our second-stage refinement module.
>
> On the other hand, disabling the density sampling in the first stage of our method leads to worse results. This aligns with our analysis in the previous response to W2.3, highlighting the crucial role of this simple yet effective approach in learning better voxel density.
>
> We will include this voxel-density evaluation in the revised version. It was your reminder that prompted us to include this previously omitted ablation study. Thanks again!
>
> *[1] Panos Achlioptas, Olga Diamanti, Ioannis Mitliagkas, and Leonidas Guibas. Learning representations and generative models for 3d point clouds. In International conference on machine learning, pp. 40–49. PMLR, 2018.*

---

### Official Review · Reviewer_LssR · 2025-11-02

**Soundness:** 2
**Presentation:** 3
**Contribution:** 2
**Rating:** 4
**Confidence:** 3

**Summary:**

This paper introduces ResGen, a two-stage framework for generating realistic LiDAR point clouds. The first stage uses a diffusion model to generate coarse representation of point clouds in voxel. The second fine-tuning stage uses the produced Residual Diffusion Model to generate the precise representation, conditioned on the fixed coarse input. The method is evaluated unconditionally on KITTI-360, showing strong gains in JSD, MMD, and a custom density metric (REAP).

**Strengths:**

1.  **Comprehensive Experimental Evaluation**
The paper provides extensive quantitative and qualitative evaluations under both conditional and unconditional generation settings. As demonstrated in the reported metrics and visualization results, the proposed method consistently outperforms the baseline approaches in terms of generation precision.


2.  **Modular and Well-Structured Pipeline**
The framework is cleanly decomposed into a modular two-stage pipeline, as clearly illustrated in Figure 1. The data flow from 2D latent representations to the refined point cloud is logically presented, and the residual generation mechanism offers an intuitive and effective refinement strategy.

**Weaknesses:**

1.  **Insufficient Evaluation on benchmarks and baselines**
I. The quantitative comparisons are primarily conducted on unconditional generation using the KITTI-360 dataset. However, the quantitative experimtnes of the text-conditioned point cloud generation task is missing.
II. All compared baseline methods were published prior to 2025. Including more recent state-of-the-art approaches would strengthen the experimental validity and better situate the proposed method within the current research landscape.

2.  **Moderate Technical Novelty**
    While the coarse-to-fine framework and residual learning mechanism are intuitive and effective, they have been widely explored in other computer vision domains, such as LiDAR odometry and scene flow estimation. Furthermore, the strategy of “starting from a given prior sample,” though practical, resembles a simple yet effective training trick seen in prior works (e.g., ADD [1]).

3.  **Insufficient Failure and Diversity Analysis**
    The paper does not provide an analysis of failure cases or a systematic evaluation of output diversity. These aspects are crucial for thoroughly assessing the robustness and generative performance of the proposed method.


[1] Sauer A, Lorenz D, Blattmann A, et al. Adversarial diffusion distillation[C]//European Conference on Computer Vision. Cham: Springer Nature Switzerland, 2024: 87-103.

**Questions:**

Given the relatively intuitive pipeline of the proposed method, the experimental results carry significant weight in demonstrating its overall effectiveness and contributions. In addition to addressing the weaknesses previously outlined, the following issues also require attention:

1.There are notable inconsistencies between the performance metrics of baseline methods as reported in this paper and their original publications. For instance, the JSD score for LiDM on unconditional generation is reported as 0.439 here, whereas it was 0.211 in the original work. The authors should clarify the source of these discrepancies—whether they stem from differences in implementation, evaluation protocols, or data processing.

2.As shown in Figure 6, the proposed method generates point clouds with notably sharper edges compared to the curved and blurred outputs of LiDiff. Is this attributed to the use of voxel representation towards the point cloud structure? It raises a further question: why is this characteristic not observed in the results of other methods presented in Figure 4?

---

> ### Author Response · Authors · 2025-11-21
> **Response to Reviewer LssR (Part 1/2)**
>
> We sincerely appreciate your careful reading of our manuscript and the questions and suggestions you raised, which are really insightful and targeted. We highly value your feedback and will provide detailed responses to your concerns.
>
> ### *W1.1: Quantitative results for text conditioned generation*
>
> **We believe that establishing evaluation metrics for text-controlled LiDAR point cloud generation is itself a scientific challenge.** In the image domain, metrics like CLIP Score are widely used for text-conditioned generation assessment. However, in the point cloud domain, especially for driving scenes, no foundation model like CLIP currently exists. We plan to address this critical gap in our future work and sincerely appreciate you raising this important point.
>
> ### *W1.2: Comparison with recent state-of-the-art approaches*
>
> Thank you very much for your suggestion, and we apologize for our oversight. For the single-frame point cloud generation, we conducted additional comparisons with R2Flow [1] (published at ICRA 2025) and SG-LDM [2] (published at ICCV 2025). For SG-LDM, we used its unconditional model (guidance scale = 0). Since it was developed on the SemanticKITTI dataset and no training code was released, we adapted our method to the SemanticKITTI dataset for a fair comparison.
>
> For the multi-frame accumulated scenario, we compared with 3DiSS [3], a 2025 arXiv preprint. Given that 3DiSS performs generation at the occupancy level, we voxelized our point cloud results using the same grid size to ensure a fair comparison.
>
> We sampled 200 instances from each method for evaluation. All results were generated using the officially released weights for inference. The experimental results are presented below. Following our paper, **REAP** is used to measure the density realism of point clouds, indicating the relative error between the average point counts of generated samples and real-world samples.
>
> 1. Comparison with R2Flow on KITTI-360
>
> | Method | JSD | MMD(e-2) | REAP(%) |
> | --- | --- | --- | --- |
> | **Ours** | **0.253** | **0.248** | **1.71** |
> | R2Flow | 0.431 | 0.251 | 41.15 |
>
> 2. Comparison with SG-LDM on single-frame SemanticKITTI
>
> | Method | JSD | MMD(e-2) | REAP(%) |
> | --- | --- | --- | --- |
> | **Ours** | **0.224** | **0.200** | **7.68** |
> | SG-LDM | 0.418 | 0.211 | 58.78 |
>
> 3. Comparison with 3DiSS on **voxelized,** multi-frame accumulated SemanticKITTI
>
> | Method | JSD | MMD(e-2) |
> | --- | --- | --- |
> | **Ours** | **0.510** | **0.172** |
> | 3DiSS | 0.734 | 0.173 |
>
> The results above demonstrate that our method maintains an advantage when compared against the latest state-of-the-art approaches.
>
> **The revised version of the paper includes figures showcasing visual comparisons.** As shown in Figure 10 in Appendix D of the paper, our method generates more stable structures and more reasonable layouts compared with R2Flow.
>
> ### *W2: Uniqueness of our residual learning and coarse-to-fine framework*
>
> We humbly emphasize that both the motivation behind our Residual Model and its core insights are distinct from existing approaches, and our two-stage framework offers unique advantages compared to other methods.
>
> **Residual Model** While residual learning has been widely explored, the motivation and core insight behind each approach are fundamentally different. For instance, many methods introduce residual learning primarily to make the network easier to train. In contrast, the **key motivation in our approach is to address the theoretical limitation** in existing methods—they employ an incomplete diffusion process, which implicitly assumes a Gaussian distribution that does not hold in many cases. Starting by identifying this limitation, *we provide a comprehensive theoretical and experimental analysis.* We expect that this analysis can offer a new insight for the design of future point cloud diffusion models. Under this theoretical framework, a series of specific implementations can be derived, which is also one of the focuses of our next steps.
>
> **Two-stage framework** Our two-stage framework **offers better universality compared to previous methods**. Existing range image based approaches are limited to single-frame point clouds obtained from mechanical LiDAR scans and require knowledge of the LiDAR hardware parameters. In contrast, our method not only generates single-frame point clouds for both mechanical and solid-state LiDAR, but also handles multi-frame accumulated point clouds. Moreover, our coarse generation essentially constitutes a density field, which adapt well to the non-uniform nature of point clouds. Subsequently, building upon the current method to develop a unified point cloud generation framework across scales and types is also one of the directions for our future work.

---

> ### Author Response · Authors · 2025-11-21
> **Response to Reviewer LssR (Part 2/2)**
>
> ### *W3: Diversity and failure case analysis*
>
> Thank you for your highly professional suggestions, and we apologize for overlooking these two points. We also strongly agree that failure cases are important. Overall, the limitations of our method primarily lie in two aspects: occasional local artifacts and constrained numerical control.
>
> **One limitation of our generation method, also a common limitation of current methods,  is that it occasionally produces some blurred outliers or noise points**, as shown in Figure 11 in Appendix D of the revised paper.
>
> This common limitation is also pointed out by R2Flow [1]. However, our method has already shown improvement in this aspect compared to others. As shown in **Figure 11 in Appendix D** of the revised paper, previous methods generate more blurred content.
>
> **Another limitation of our method is that it currently cannot achieve precise numerical control** over generated attributes, such as the exact width of roads or the specific distance to vehicles. We recognize this as a fundamental limitation of text-based conditioning. To address this, we plan to incorporate layout-based control conditions in future work. Compared to the relatively under-explored area of text-guided generation, layout conditioning has been more extensively studied. Intuitively, in our method, since the control condition (layout) and the generation target of our first step (BEV latent) share a similar spatial modality, the implementation is more straightforward.
>
> **For diversity analysis, our method demonstrates good diversity, generating scenes that encompass various types of roads**: rural or urban, with or without vehicles, wide or narrow, straight roads or intersections, etc. Beyond Figure 4 and Figure 6 in the paper, **Figure 12 in Appendix D of the revised paper** provides additional examples of generated results.
>
> ### *Q1: Evaluation details*
> Thank you for such a thorough review! We apologize for the confusion caused. The difference in results stems from evaluation protocols. Our evaluation adopts a bounded 50-meter square space and a smaller voxel size of 0.03m to better reflect the fine-grained details in the JSD metric, avoiding destroying the “ring-like pattern” of LiDAR scanning in short-range regions. For MMD, unlike LIDM which employs a 2D projection MMD, we utilized 3D MMD to better measure geometric similarity.
>
> ### *Q2: The reason of sharper edges, and why not reflected in Figure 4*
>
> Thank you for your careful review. Actually, the characteristic of flying blurred edges can also be observed in Figure 4 of the paper, and we add **Figure 13 in Appendix D** to highlight the characteristic. We humbly clarify that the following reason may cause the reviewer’s potential misunderstanding:  In Figure 6, we use the point completion method LiDiff to convert the single-frame results (eg., the ones in Figure 4) to the multi-frame accumulated version, which significantly amplifies the visual effect of “blurred edges” by densifying the flying edge points. For better illustration, we add **Figure 14 in Appendix D** to demonstrate such amplification.
>
> *[1] Kazuto Nakashima, Xiaowen Liu, Tomoya Miyawaki, Yumi Iwashita, and Ryo Kurazume. Fast lidar data generation with rectified flows. In 2025 IEEE International Conference on Robotics and Automation (ICRA), pp. 10057–10063. IEEE, 2025.*
>
> *[2] Zhengkang Xiang, Zizhao Li, Amir Khodabandeh, and Kourosh Khoshelham. Sg-ldm: Semantic-guided lidar generation via latent-aligned diffusion. In Proceedings of the IEEE/CVF International Conference on Computer Vision (ICCV), pp. 24965–24976, October 2025.*
>
> *[3] Lucas Nunes, Rodrigo Marcuzzi, Jens Behley, and Cyrill Stachniss. Towards generating realistic 3d semantic training data for autonomous driving. arXiv preprint arXiv:2503.21449, 2025.*

---

> ### Author Response · Authors · 2025-11-27
> **Response to Reviewer LssR (Additional Visualizations)**
>
> Dear Reviewer LssR,
>
> We sincerely thank you again for your highly professional review comments. To better address your concern regarding the **insufficient analysis of generation diversity**, we supplemented our previous response (which added 8 visualization samples) with **more visualizations in Figure 15 of the latest revised paper**. We randomly selected **30 samples** generated by our method, with 15 samples each from the single-frame and multi-frame accumulated scenarios, hoping to more comprehensively address your concern. We look forward to your feedback. Thank you!

---

### Official Review · Reviewer_dBaQ · 2025-11-02

**Soundness:** 3
**Presentation:** 3
**Contribution:** 3
**Rating:** 6
**Confidence:** 3

**Summary:**

This paper proposes ResGen, a novel two-stage framework for high-fidelity LiDAR-based point cloud generation, aiming to overcome the limitations of resolution in occupancy methods and applicability in range image methods. The method first uses a 3D VAE on density voxels combined with a 2D Latent Diffusion Model to generate a coarse 3D structure. The core innovation lies in the second stage: a Residual Diffusion Model (Residual DM) for refinement. Motivated by a pilot study that rejects the commonly implicit assumption of Gaussian residuals in previous "Partial Diffusion Models" (e.g., LiDiff), the authors propose to directly model the diffusion process for the residual $\delta = x_{tar} - x_{pri}$, conditioned on the coarse prior $x_{pri}$. This approach is shown to be theoretically sounder and empirically superior. Experiments on KITTI-360 and SemanticKITTI demonstrate state-of-the-art results across various metrics (JSD, MMD, REAP, User Preference) for both single-frame and multi-frame accumulated point cloud generation.

**Strengths:**

1. The most significant contribution is the rigorous analysis and subsequent proposal of the Residual Diffusion Model.
2. ResGen achieves superior quantitative results compared to all tested baselines on unconditional generation (Table 1).
3. The coarse-to-fine pipeline effectively addresses the challenge of generating large-scale, high-resolution 3D data by separating structural generation from detail refinement.

**Weaknesses:**

1. The VAE encoder transforms 3D density voxels into 2D feature maps by "simply merged" the information along the Z-dimension (lines 147-151). While the authors state this is sufficient for the coarse stage, this process inherently discards or compresses depth/height information into a 2D plane, which could limit the quality of the initial structural prior. It would be beneficial to provide more detail on this merging operation.
2. In Table 3, the two variants of the Residual DM—Residual only and Concatenated show almost same performance. This suggests that using one over the other provides no practical benefit. This finding should be highlighted more strongly as it might simplifies the network.

**Questions:**

1. You trained LDM, which mean we are not working with direct voxels. does the data is not normal distributed in the latent as well?
2. This all method is build on diffusion, but the world shifted to flow matching, does your method can be shift as well?
3. Please provide the explicit mathematical operation used to "simply merge" the information along the Z-dimension of the 3D feature voxels into 2D feature maps in the VAE encoder

---

> ### Author Response · Authors · 2025-11-21
> **Response to Reviewer dBaQ (Part 1/2)**
>
> Thank you very much for your positive remarks and constructive comments on our methodology. It has been highly beneficial in guiding us toward further improvements.
>
> ### *W1&Q3: “Simple merging operation” in the VAE*
>
> **Our simple merging operation refers to treating the height dimension as the feature dimension for 2D convolution.**
>
> **Mathematical expression** As shown in the VAE encoder in Figure 1 (D) of the paper, we begin with a 3D feature tensor $\mathbf{F}_{3D} \in \mathbb{R}^{f \times X \times Y \times Z}$, where $f$ is the number of feature channels, and $X$, $Y$, $Z$ are the spatial dimensions. Our merging operation first transpose the tensor into $\mathbf{F}'\_{3D} \in \mathbb{R}^{f \times Z \times X \times Y}$, and then flatten the height ($Z$) dimension into the channel dimension to obtain a 2D feature tensor  $\mathbf{F}\_{2D} \in \mathbb{R}^{(f \cdot Z) \times X \times Y}$.
>
> Note that this rearrangement operation **does not lose height information.** It is not a kind of “compression”, as it merely changes the way the data is interpreted.
>
> Thank you very much for your comment. We also believe that increasing model capacity would benefit performance, and we plan to explore this in our future work. Given the extensive existing research in this area, particularly within BEV-based point cloud detection, we will conduct a thorough survey to inform our approach.
>
> ### *W2: Treatment of the almost same performance of  two Residual DM varients*
>
> Thank you very much for your suggestion. **This is indeed a good point to demonstrate the effectiveness of our method**, which we had not previously given sufficient attention. In the Concatenated implementation, the feature dimensionality is increased compared to the original LiDiff, thereby also increasing network complexity. Therefore, its performance improvement cannot be entirely attributed to our Residual Diffusion Model. In contrast, the Residual Only approach maintains the same feature dimensionality and computational cost as the original method, making it clearer evidence that the performance gain stems from the construction of our Residual Diffusion Model. We will include this evaluation in the revised version.
>
> ### *Q1: Distribution of the latent space*
>
> Thank you for your question. To answer your question, we randomly selected 2,000 samples from the KITTI dataset and encoded them using our trained VAE encoder to obtain latent features. We plot the distribution of each dimension of the features.
>
> **Results are shown in Figure 9 of Appendix D of the revised paper**, where the green histogram represents the empirical distribution, and the black curve represents the normal distribution fitted using the mean and variance of the green distribution.
>
> As shown, the overall distribution approximates a Gaussian shape, which aligns with statistical expectations. However, we also observe that in many dimensions, the probability of values near zero is higher than what the fitted Gaussian curve would suggest. Considering that zero often serves as a "transition point" in neural network activation functions, small variations in feature values around this point can lead to significant changes in the network's output. Therefore, having feature values concentrated near such critical points enhances the network's capacity to generate diverse outputs. This, in a way, reflects the effectiveness of our feature learning.
>
> Additionally, we would like to clarify that **the Gaussian distribution assumption discussed in our paper focuses on the second-stage** point cloud diffusion model, not the first-stage latent space diffusion model.

---

> ### Author Response · Authors · 2025-11-21
> **Response to Reviewer dBaQ (Part 2/2)**
>
> ### *Q2: Shift to flow matching*
>
> Yes, in our method, both the 2D latent diffusion in the first step and the 3D point cloud diffusion in the second step can be replaced by flow matching.
>
> **Compatibility with Our Residual Model** Specifically, there is no theoretical gap when shifting the proposed Residual Diffusion Model to a “Residual Flow Matching Model”. We demonstrate this as follows.
>
> Formally, given the target residual distribution $p\_{\text{res}}(\mathbf{\boldsymbol{\delta}})$ and a base distribution $p\_0(\mathbf{\boldsymbol{\delta}})\sim \mathcal{N}(0, \mathbf{I})$, a **Resiudal Flow Matching Model** seeks a time-dependent vector field $\mathbf{v}\_t(\mathbf{\boldsymbol{\delta}})$ such that the solution to the ordinary differential equation (ODE) $$ \frac{d\mathbf{\boldsymbol{\delta}}\_t}{dt} = \mathbf{v}\_t(\mathbf{\boldsymbol{\delta}}\_t;\textbf{x}\_\text{pri}) $$
> transports samples from $p\_0$ to $p\_{\text{res}}$ as $t$ evolves from $0$ to $1$.
> Here we explicitly write $\textbf{x}\_\text{pri}$, a constant condition of the field, for clarity.
>
> Take the optimal transport conditional vector field as example, where
> the vector field has the form $u\_t(\boldsymbol{\delta}|\boldsymbol{\delta}\_1)= \frac{\boldsymbol{\delta}\_1-(1-\sigma\_{min}) \boldsymbol{\delta}}{1-(1-\sigma\_{min}) t}$.
>
> Corresponding **Training Objective**:
> Given a pair of samples $\mathbf{\boldsymbol{\delta}}\_0 \sim p\_0$ and $\mathbf{\boldsymbol{\delta}}\_1 \sim p\_{\text{res}}$, the model $\mathbf{v}\_\theta$ is trained by minimizing $$ \mathcal{L}(\theta) = \mathbb{E}\_{\mathbf{\boldsymbol{\delta}}\_0, \mathbf{\boldsymbol{\delta}}1, t} \Big\| \mathbf{v}\_\theta(\mathbf{\boldsymbol{\delta}}\_t, t; \textbf{x}\_\text{pri}) - (\mathbf{\boldsymbol{\delta}}\_1 - (1-\sigma\_{min})\mathbf{\boldsymbol{\delta}}\_0) \Big\|^2. $$
>
> **Inference Formula**:
> To generate a sample, solve the ODE $$ \frac{d\mathbf{\boldsymbol{\delta}}t}{dt} = \mathbf{v}\_\theta(\mathbf{\boldsymbol{\delta}}\_t, t; \textbf{x}\_\text{pri}) $$ with initial condition $\mathbf{\boldsymbol{\delta}}\_0 \sim p\_0$ and integrate from $t=0$ to $t=1$ to obtain the targeted residual $\mathbf{\boldsymbol{\delta}}\_1$.

---

> ### Author Response · Authors · 2025-11-25
> **Response to Reviewer dBaQ (Part 2 Addition: Experimental Results)**
>
> **Experiment Results** Based on the theoretical analysis in **Part 2** (https://openreview.net/forum?id=v1gaXJDCbf&noteId=wnqStYTzYX), we conducted experiments, with results presented in the table below. We sampled 200 instances for evaluation.
>
> | Method | JSD | MMD(e-2) |
> | --- | --- | --- |
> | Ours w/o Residual generation | 0.738 | 2.483 |
> | Ours w/ Residual Diffusion Model | 0.253 | 2.483 |
> | Ours w/ Residual Flow Matching | **0.245** | **2.481** |
>
> As shown, in the second residual generation step of our method, **replacing the diffusion-based implementation with flow matching leads to further performance improvements**. We will include these updated results in the revised version of our paper. Thank you for your valuable suggestion, which directly contributed to enhancing the effectiveness of our method!

---

### Author Response · Authors · 2025-11-27
**We are open to further discussion**

We extend our sincere gratitude to all the reviewers for your careful and insightful feedback. Your comments have been extremely valuable in helping us refine our work and have provided us with clear direction for improvement. We warmly welcome any further discussion and would be more than willing to provide responses to any additional comments. Please accept our deepest appreciation once again.

---

### Author Response · Authors · 2025-12-03
**Rebuttal Summary**

Dear Area Chair,

We sincerely thank you for taking the time to review our work. For your convenience, we have summarized the reviewers' comments and our main responses below. We received **three reviews** with the ratings of **6, 4, and 6**. We would like to express our gratitude once again to all reviewers for their professional and helpful comments.

### *Reviewer dBaQ (Rating: 6)*

Reviewer dBaQ recognizes our **rigorous analysis** of the residual and the **effectiveness** of our coarse-to-fine pipeline.

Comments & Responses:

- Regarding the implementation of the merging operation, we added an explanation, clarifying that this operation **does not lose information**. *(W1&Q3)*
- As suggested, we **added visualizations** for the distribution of our encoded latent space. *(Q1)*
- Following the suggestion to shift to flow matching, we **conducted experiments** to verify compatibility and achieved improved results. *(Q2)*

### *Reviewer LssR (Rating: 4)*

Reviewer LssR acknowledges our **comprehensive** experimental evaluation and our **clean, structured** framework.

Comments & Responses:

- Considering more recent state-of-the-art methods, we **added comparisons** with three works from ICCV 2025, ICRA 2025, and ArXiv 2025. The results confirm that our method **remains superior**. *(W1.2)*
- Regarding the uniqueness of our coarse-to-fine framework and residual learning, we clarified that our uniqueness lies in four aspects: **rigorous theoretical analysis** of the residual; a coarse density field resolving the challenge of **adaptive** point sampling; distinct **motivation**; high **universality**. *(W2)*
- As suggested, we added failure case analysis and **over 30 visualizations** to provide a more comprehensive assessment of our method's diversity. *(W3)*
- For the confusion regarding the metrics, we clarified the evaluation protocol. *(Q1)*

### *Reviewer Qfi7 (Rating: 6)*

Reviewer Qfi7 acknowledges our **novel** generation procedure and **superior** performance.

Comments & Responses:

- Regarding why our voxel generation is non-trivial, we further explained that our voxels **essentially constitute a density field**, which adapts well to the non-uniform nature of point clouds. *(W1)*
- To further clarify the training process of multi-frame scene generation, we emphasized that it **follows the same pipeline** as single-frame generation, highlighting the generalizability of our method. *(W2.4)*
- To further clarify the user study, we **provided a detailed description** of its design. *(Q1)*
- As suggested, we added a voxel-level evaluation, which offers a novel perspective to **demonstrate the effectiveness** of our method. *(Q2)*

We hope that our summary will assist you in the evaluation process. Once again, we deeply appreciate your time and effort in reviewing our work, and we eagerly look forward to your valuable feedback.

---

### Meta-Review · Area_Chair_ruzk · 2026-01-07

**Summary:**

Two major concerns are raised:
- Novelty. The proposed corase-to-fine two-stage framework works by first generating a coarse 3D structure and then refines the structure to dense point clouds via a proposed residual diffusion model. However, similar approaches and methods are already witnessed and applied in many other domains. There also lacks evidences showing the effectiveness of the proposed residual diffusion model.
- Insufficient evaluation and analysis of the experimental results. The reviewers pointed out multiple missing experiments, e.g., ablation study on the voxel density, analysis of failure cases, etc.

**Reviewer Concerns:**

The concerns regarding the experimental results are mostly addressed.

**Reviewer Scores:**

No change of score is expected.

---

### Decision · Program_Chairs · 2026-01-26

Reject